# Survival Outcomes in Older Women with Oestrogen-Receptor-Positive Early-Stage Breast Cancer: Primary Endocrine Therapy vs. Surgery by Comorbidity and Frailty Levels

**DOI:** 10.3390/cancers16040749

**Published:** 2024-02-11

**Authors:** Yubo Wang, Douglas Steinke, Sean P. Gavan, Teng-Chou Chen, Matthew J. Carr, Darren M. Ashcroft, Kwok-Leung Cheung, Li-Chia Chen

**Affiliations:** 1Centre for Pharmacoepidemiology and Drug Safety, Division of Pharmacy and Optometry, School of Health Sciences, Faculty of Biology, Medicine and Health, The University of Manchester, Manchester Academic Health Science Centre, Stopford Building, Oxford Road, Manchester M13 9PT, UK; douglas.steinke@manchester.ac.uk (D.S.); teng-chou.chen@manchester.ac.uk (T.-C.C.); matthew.carr@manchester.ac.uk (M.J.C.); darren.ashcroft@manchester.ac.uk (D.M.A.); li-chia.chen@manchester.ac.uk (L.-C.C.); 2Manchester Centre for Health Economics, Faculty of Biology, Medicine and Health, The University of Manchester, Oxford Road, Manchester M13 9PL, UK; sean.gavan@manchester.ac.uk; 3NIHR Greater Manchester Patient Safety Research Collaboration (PSRC), The University of Manchester, Manchester M13 9PT, UK; 4Royal Derby Hospital Centre, School of Medicine, University of Nottingham, Uttoxeter Road, Derby DE22 3DT, UK; kwok_leung.cheung@nottingham.ac.uk

**Keywords:** comorbidities, early-stage breast cancer, frailty, older postmenopausal women, primary endocrine therapy, surgery

## Abstract

**Simple Summary:**

This study investigated the outcomes of primary endocrine therapy (PET) compared to surgery for treating older women (≥70 years) with early-stage oestrogen-receptor-positive breast cancer, specifically considering the impact of patients’ frailty and comorbidity levels. Using UK data from 2000 to 2016, the researchers analysed the all-cause mortality and breast cancer-specific mortality among 23,109 patients, stratified by their Charlson comorbidity index and hospital frailty risk score. The findings showed that surgery generally resulted in better overall survival rates compared to PET in patients with higher levels of frailty or comorbidity. Notably, in very frail older women, there was no significant difference in breast cancer-specific mortality between PET and surgery arms. The study concludes that PET could be a viable treatment option for frail older women with early-stage breast cancer, considering the minimal survival benefits offered by surgery in this subgroup.

**Abstract:**

Primary endocrine therapy (PET) offers non-surgical treatment for older women with early-stage breast cancer who are unsuitable for surgery due to frailty or comorbidity. This research assessed all-cause and breast cancer-specific mortality of PET vs. surgery in older women (≥70 years) with oestrogen-receptor-positive early-stage breast cancer by frailty and comorbidity levels. This study used UK secondary data to analyse older female patients from 2000 to 2016. Patients were censored until 31 May 2019 and grouped by the Charlson comorbidity index (CCI) and hospital frailty risk score (HFRS). Cox regression models compared all-cause and breast cancer-specific mortality between PET and surgery within each group, adjusting for patient preferences and covariates. Sensitivity analyses accounted for competing risks. There were 23,109 patients included. The hazard ratio (HR) comparing PET to surgery for overall survival decreased significantly from 2.1 (95%CI: 2.0, 2.2) to 1.2 (95%CI: 1.1, 1.5) with increasing HFRS and from 2.1 (95%CI: 2.0, 2.2) to 1.4 (95%CI 1.2, 1.7) with rising CCI. However, there was no difference in BCSM for frail older women (HR: 1.2; 0.9, 1.9). There were no differences in competing risk profiles between other causes of death and breast cancer-specific mortality with PET versus surgery, with a subdistribution hazard ratio of 1.1 (0.9, 1.4) for high-level HFRS (*p* = 0.261) and CCI (*p* = 0.093). Given limited survival gains from surgery for older patients, PET shows potential as an effective option for frail older women with early-stage breast cancer. Despite surgery outperforming PET, surgery loses its edge as frailty increases, with negligible differences in the very frail.

## 1. Introduction

Breast cancer is increasingly diagnosed at an older age. In the United Kingdom (UK), early-stage breast cancer is predominately diagnosed in postmenopausal women (86% aged ≥ 50 years), but 46% are at older ages (age ≥ 70 years) [1]. The optimisation of treatments for older female patients with breast cancer presents challenges due to the scarcity of robust evidence in treatment effectiveness [2,3], constrained clinical capacity for managing complex cases [4], and the preferences of patients for less invasive treatments [5]. 

According to the national clinical guideline in England, irrespective of a patient’s age at diagnosis, surgery and adjuvant systemic therapy are recommended as the first-line treatment for operable breast cancer that is oestrogen receptor positive (ER+) unless significant comorbidity precludes surgery [6]. Evidence has shown that removing the tumour by surgery significantly extends the survival of postmenopausal women with early-stage operable breast cancer [7]. Nevertheless, among older female patients deemed unsuitable for surgical intervention, there exists a subset whose life expectancy might be diminished or whose prospects for survival might not be optimally realised through surgery, primarily owing to their compromised physical condition characterised by comorbidities and frailty [4]. Consequently, this cohort is experiencing a growing shift towards non-surgical therapeutic options [8].

Primary endocrine therapy (PET) has been considered an alternative to surgery for older female patients diagnosed with ER+ early-stage breast cancer, particularly those for whom surgery is unsuitable [8]. Nevertheless, its comparative effectiveness has not been comprehensively studied. In the most recent meta-analysis encompassing seven randomised controlled trials (RCTs) and six non-RCTs, a comparative evaluation was conducted between 2829 cases subjected to PET and 11,425 cases subjected to surgical intervention [9]. The findings elucidated that while PET exhibited non-inferior regarding overall survival compared to surgery, it was non-inferior concerning breast cancer-specific mortality [9].

However, these results are not seen in clinical practice as physically fit patients are seldom offered PET [10]. Besides those RCTs, the lone RCT conducted by Reed et al. in 2008, which sought to compare primary endocrine therapy with surgical intervention in older women diagnosed with early-stage breast cancer by various health statuses, was hampered by its inability to recruit an adequate number of participants [11]. Meanwhile, patients with multi-morbidity or frailty are often excluded from RCTs due to the risk of serious adverse events [12,13]; thus, treatment effectiveness in those vulnerable patients is scarce.

Due to the challenge of conducting RCTs in older women with physical functioning impairment, observational studies are considered a feasible approach to evaluate the effectiveness of PET in older women. In line with findings from earlier RCTs, several observational studies investigating older women with breast cancer and impaired physical functioning have reported better overall survival in patients undergoing surgery than those undergoing PET [14,15,16,17,18,19]. However, these findings are susceptible to confounding by indication, presenting a challenging bias to overcome. Additionally, given the presence of frailty or comorbidity in this patient population, there may be an increased risk of non-cancer mortality, leading to higher ’all-cause mortality’ (poor overall survival), a commonly used outcome measure in oncology [20]. Consequently, the comparative effectiveness of PET versus surgery in older female patients with various degrees of physical functioning impairment remains inconclusive.

To address this knowledge gap, this study aimed to compare all-cause (ACM) and breast cancer-specific mortality (BCSM) associated after PET and surgery in older women (aged ≥ 70 years) with newly diagnosed early-stage oestrogen-receptor-positive breast cancer at different frailty and comorbidity levels. Rigorous examination and sensitivity analysis were conducted to inform appropriate treatment approaches for this patient cohort.

## 2. Methods

This study presents a cohort study reported according to the Strengthening the Reporting of Observational Studies in Epidemiology (STROBE) statement (Appendix A). A diagram of this cohort study design is provided in Figure 1. 

### 2.1. Data Source

This study used anonymous primary care electronic health records from CPRD Gold and Aurum, linked with three national data sources: (1) Hospital Episode Statistics (HES) subsets, including Admitted Patient Care (APC), Outpatient (OP), and Accident and Emergency (AE) records; (2) Cancer Registrations from NCRAS; and (3) patient-level Death Registration and Index of Multiple Deprivation (IMD) from ONS (Table 1).

The study covered the period from January 2000 to May 2019, chosen to minimise potential COVID-19 impact on geriatric cancer treatment and coincide with comprehensive ONS Death Registration data (Table 1). CPRD Gold and Aurum, HES, NCRAS, and ONS Death Registration data were extracted and processed by the CPRD Centre. NCRAS records were linked and extracted by Public Health England. The study protocol received approval from the Independent Scientific Advisory Committee (ISAC) in April 2020 (protocol ID: 20_079R).

### 2.2. Study Population

A dynamic cohort comprised older women (≥70 years at diagnosis) newly diagnosed with ER+ early-stage breast cancer was identified from people registered in the general practices from CPRD Gold and Aurum and the linkage to HES, NCRAS, and ONS Death Registration from 2000 to 2016 (Table 1). The inclusion duration was the maximal overlapped period among all databases (Table 1). During the inclusion period, the study cohort fulfilled the following inclusion criteria: (1) female patients having records in CPRD, NCRAS and HES databases; (2) had a record of newly-diagnosed breast cancer, either ICD-10 code (C50) in HES, and Read or EMIS codes in CPRD Gold or Aurum; (3) aged ≥ 70 years at the diagnosis of breast cancer; (4) diagnosed with ER+ early-stage (stage I, II, or IIIA) breast cancer; and (5) had up-to-standard practice data in the CPRD. 

The code list for breast cancer (as a subset of the 20 site-specific cancers) was developed by referring to the published literature [21] and an algorithm generated by the research team (Appendix A). Early-stage breast cancer was defined according to the stage (stage I, II, and IIIA) recorded in NCRAS, which can undergo surgery as primary treatment. If the stage or ER status record was missing, the initial treatment was used as a proxy to fill in the missing data (i.e., if patients were administered endocrine therapy), as it is imperative that patients possessed an ER+ status. All patients with surgery, regardless of combining endocrine therapy as the initial treatment, were identified as early-stage breast cancer [6]. General practices in CPRD were classified as ‘up to standard’ when the practice met specified minimum quality criteria [21,22,23,24].

In addition, individuals were excluded if they had a “Death Certificate Only” flagged in the NCRAS cancer registry, which means breast cancer diagnosis was retrieved from the death certificate; thus, the accurate date of diagnosis was unavailable. Furthermore, patients with the same diagnosis and death date were excluded as the follow-up time was unavailable. The study cohort was followed up from the diagnosis date (index date) to the study endpoint, i.e., the date of death, transfer out of practice, or the end of the follow-up (31 May 2019), whichever appeared first.

### 2.3. Treatment Exposure

Patients’ initial breast cancer treatments, received within 12 months of diagnosis, were divided into surgery and PET groups. The surgery group encompassed breast-conserving surgery (BCS) or mastectomy, with or without neoadjuvant or/and postoperative adjuvant therapies, e.g., chemotherapy, radiotherapy, or endocrine therapy. The PET group included patients who did not undergo surgery but received endocrine therapy, possibly combined with other treatments within the 12-month window in line with the clinical practice quality assessment [6,25].

Initial treatments were primarily identified through NCRAS Cancer Registration, complemented by surgical procedure codes (OPCS-4) in the HES APC database or endocrine therapy codes (BNF) in the CPRD Gold and Aurum databases. Inconsistent surgery dates between NCRAS and HES were resolved by selecting the earlier recorded date (Appendix A).

### 2.4. Outcome Measure

Overall survival (OS), estimated from all-cause mortality (ACM) events, was the primary outcome of this study. In addition, breast cancer-specific mortality (BCSM) was measured as a secondary outcome as it minimises confounding of other competing risks (e.g., ageing, frailty, or multi-morbidities) leading to deaths [20]. ACM events were identified from the ONS death registration record, and BCSM events were identified by screening the ICD-10 codes (C50.X) on ONS death registration [26].

### 2.5. Covariates

Patients’ baseline characteristics that may influence mortality risk, including age at diagnosis (on the index date), socioeconomic status, comorbidities, and frailty, were identified up to one year before the index date as covariates [27,28,29]. Nineteen comorbidities, identified by Read and SNOMED or EMIS codes in CPRD Gold and Aurum and ICD-10 codes in HES, were used to calculate the Charlson comorbidity index (CCI) using validated algorithms [30] (Appendix A). The calculated CCI was further stratified into three levels: low (0–2), intermediate (3–4), and high (≥5) [30].

Symptoms related to frailty (n = 109) identified by screening ICD-10 codes in HES were used to calculate the hospital frailty risk score (HFRS) [31] (Appendix A). Similarly, the estimated HFRS was then stratified into three levels: non-frail (<5), pre-frail (5–15), and frail (≥15) [31]. The IMD, recorded as a decile from 1 (most deprived area) to 10 (least deprived area) in ONS data, was grouped into five categories (I: 1–2; II: 3–4; III: 5–6; IV: 7–8; V: 9–10) to indicate socioeconomic status.

### 2.6. Data Analysis

Descriptive statistics were used to report the characteristics of patients, exposure variables, outcome variables, and covariates. The Chi-square and Wilcoxon signed-rank tests assessed whether differences in categorical or continuous variables, respectively, were statistically significant (*p* < 0.05). Kaplan–Meier (KM) survival curves were used to present the survival outcomes by treatment and levels of CCI and HFRS.

The propensity score was estimated, reflecting the likelihood of patients opting for surgery or PET, considering four confounders: age (in 5-year intervals), three-level HFRS and CCI, and five-level IMD [32], which influence initial breast cancer treatment choices [33]. Patients were then assigned weights equal to the inverse of their predicted surgery probability, forming inverse probability treatment weights (IPTW) based on the propensity score [33]. We examined the standardised mean difference (SMD) to assess the balance between treatment groups. An SMD below 0.1 signified a well-balanced covariate distribution across both treatment groups [34].

Survival analysis was performed to compare the cumulative OS and BCSM probabilities between PET and surgery in patients with different levels of HFRS. The survival analysis considered the IPTW (weights to diminish the confounding of exposures), propensity score, and all covariates (age, levels of CCI, HFRS, and IMD) in regression adjustment as the doubly robust estimation to minimise the bias by indication [35]. Cox proportional hazard (PH) regression was used to estimate the hazard ratio (HR) and 95% confidence interval (95%CI) for ACM and BCSM between surgery versus PET at different levels of HFRS and CCI.

The proportional hazard assumption of the Cox regression was tested using the Schoenfeld residuals test, which tests the independence between residual and time [36]. In the Schoenfeld residuals test, a *p* < 0.05 implies that covariates violate the proportional hazard assumption, i.e., the hazard ratio of each category within each covariate compared to the reference group changes with time [36,37]. If the *p* < 0.05 in the Schoenfeld residuals test, then a time-varying Cox PH regression was performed. 

Furthermore, as the deaths of patients with high HFRS and CCI levels are highly associated with competing events such as cardiovascular or cerebrovascular events, a competing risk regression was conducted [38] as a sensitivity analysis to compare cause-specific hazards of different event types, considering both event rates and the influence of competing events [2]. Two competing risk regression models were evaluated: (1) assessing competing risks between ACM (censoring event) and BCSM (competing event) to check consistency with Cox PH regression; (2) exploring competing risks between other causes of death (defined as non-breast cancer-specific death) and BCSM while considering frailty and comorbidity levels. In both models, subdistribution hazard ratios (SHRs) for all covariates were estimated, and PET versus surgery was compared across three HFRS and CCI levels. Furthermore, we conducted cumulative incidence functions (CIF) stratified by CCI and HFRS levels to examine time-varying differences in both models. All statistical analyses were performed with Stata 14.0 (Stata Corp, College Station, TX, USA).

## 3. Results

### 3.1. Cohort Characteristics

Of the 23,109 older women (aged ≥ 70 years) with ER+ early-stage breast cancer identified during the inclusion period (Figure 2), 70% (n = 16,096) received surgery and 30% (n = 7013) received PET. The majority of patients, whether undergoing surgical intervention or PET, exhibited non-frailty status (89.4%/69.6%), along with a relatively low burden of comorbidities (80.4%/70.2%) (Table 2). The number of patients with both HER-2+ and ER+ was 553 (3.43%) in the surgery group and 132 (1.88%) in the PET group (Table 3).

The proportion of patients who received PET increased with age (Figure 3). When stratifying by 5-year age intervals, the proportion of older female patients undergoing PET increased from 10% for the 70–75-year group to 73% for those aged > 90 years (Figure 3). Of the 7013 patients undergoing PET, aromatase inhibitors were the dominant medications (55.5%), including letrozole (34.9%), anastrozole (19.9%), and exemestane (0.71%). Tamoxifen was received by 39.3% of patients in the PET group. The PET medication for the remaining 5.2% of patients was unknown. 

The median follow-up time of the study cohort was 5.16 years, with an interquartile range (IQR) of 2.71–8.93 years. Median follow-up time was significantly longer for patients undergoing surgery (6.49 years; IQR: 3.72–10.16 years) than those undergoing PET (2.77 years; IQR: 1.15–5.10 years) (Table 2). Patients who received surgery had better physical functioning than those undergoing PET. The median CCI score and HFRS score were significantly lower (*p* < 0.001) in patients undergoing surgery (CCI: 2.0; IQR: 2–3; HFRS: 2.9; IQR: 1.5–5.4) than in PET (CCI: 3.0; IQR: 2–4; HFRS: 5.5; IQR: 2.3–11.2) (Table 2). After applying the inverse probability treatment weighting, the adjusted covariates (i.e., 5-year age intervals, three-level CCI, three-level HFRS, and five-level IMD) were well balanced with the SMD less than 0.1 (Table 2; details in Appendix A, tumour characteristics reported in Appendix A).

### 3.2. Overall Survival Time and Factors Associated with All-Cause Mortality

Overall, patients undergoing surgery had a significantly longer OS time (median: 8.0 years, IQR: 3.68–14.19 years) than their PET counterparts (median: 3.6 years, IQR: 1.37–7.56 years), and the survival time shortened with the increase of HFRS and CCI. The propensity score weighted and adjusted Cox proportional hazard model found that patients undergoing PET had a significantly higher risk of ACM than those undergoing surgery (HR: 1.9; 95%CI: 1.8–2.0) after adjusting for other covariates (Table 4). 

Furthermore, poorer physical functioning (i.e., CCI and HFRS) and socioeconomic status (i.e., IMD) were also significantly associated with a higher risk of ACM. The higher levels of CCI and HFRS were associated with a higher HR of ACM ranging from 1.3 to 1.8 and 1.4 to 2, respectively; various IMD levels had a stable HR (ranging from 1.0 to 1.2) (Table 4). The Schoenfeld residuals test for the covariates found that the HR of each category within each covariate compared to the reference group was constant with time (*p* > 0.05), except for age.

### 3.3. Comparative Effectiveness of PET versus Surgery by Levels of Frailty and Comorbidity

When stratifying patients by levels of frailty and comorbidity (details in Appendix A), non-frail patients who underwent surgery had a significantly longer OS time (median: 6.8 years; IQR: 4.0–10.6) compared to their matched PET counterparts (median: 3.2 years; IQR: 1.4–6.0). Similarly, among patients with a low level of CCI, those who underwent surgery had a longer OS time (median: 7.0 years, IQR: 4.0–10.9) than their counterparts who received PET (median: 3.0 years, IQR: 1.2–5.6).

Among patients with the highest level of HFRS (median: 2.0 years; IQR: 0.9–4.4 years vs. 1.7 years; IQR: 0.6–3.4 years) or the highest level of CCI (median: 3.6 years; IQR: 1.6–6.8 years vs. 2.6 years; IQR: 1.0–5.0 years), those undergoing surgery had a significantly longer OS time than those undergoing PET, and the log-rank test for OS between the surgery and the PET in both high levels of CCI and HFRS had no statistical significance (*p* < 0.001) (Figure 4a,b). The comparison of patients undergoing surgery and their PET counterparts showed a significantly lower rate of breast cancer-specific mortality (BCSM) in patients with the highest level of HFRS (*p* = 0.251 in log-rank test, Figure 4c) but not in patients with the highest level of CCI (Figure 4d).

In general, as the levels of HFRS and CCI increased, the OS HRs and BCSM rates decreased (Table 5). Specifically, as the levels of HFRS increased, the HRs for ACM comparing PET against surgery decreased. For instance, the HR was 2.1 (95%CI: 2.0–2.2) at the low level of HFRS but reduced to 1.2 (95%CI: 1.1–1.5) at the high level of HFRS. Similarly, with increasing levels of CCI, the HRs of ACM comparing PET against surgery decreased from 2.1 (95%CI: 2.0–2.2) to 1.4 (95%CI: 1.2–1.7) (Table 5). 

A similar trend was observed for the comparative effectiveness of PET versus surgery regarding BCSM. The HRs of BCSM comparing PET against surgery decreased from 3.0 (95%CI: 2.8–3.2) in patients at the low level of HFRS to 1.2 (95%CI: 0.9–1.8) at the high level of HFRS; and from 3.0 (95%CI: 2.8–3.3) at the low CCI level to 1.5 (95%CI: 1.1–2.1) at the high CCI level (Table 5). Notably, at the high level of the HFRS, there was no statistical difference between PET and surgery in BCSM (Table 5).

### 3.4. Sensitivity Analysis Results

The competing risk regression analysis demonstrated concordance with the Cox PH regression analysis results regarding the comparative effectiveness of PET versus surgery. For the competing risk of PET compared with surgery by three levels of HFRS and CCI, in both model specifications 1 and 2, the gap of comparative analysis decreased with increasing levels of HFRS and CCI (Table 6). Specifically, in patients with a high level of HFRS, the competing risk analysis showed no statistically significant difference in overall survival between surgery and PET. The SHR of ACM, considering the competition from BCSM, was 1.2 (95% CI: 1.0–1.4, *p* = 0.082) in model specification 1, consistent with the Cox PH regression result of ACM (HR: 1.2; 95%CI: 1.1–1.5, *p* = 0.006). Similarly, the SHR of BCSM, compared with the other cause of mortality, was 1.1 (95% CI: 0.9–1.4, *p* = 0.261) in model specification 2. The sensitivity analysis results again indicated that PET showed non-inferior survival consequences to surgery for patients with physical functioning impairment (i.e., CCI and HFRS). (Details of sensitivity analysis described in Appendix A).

## 4. Discussion

Surgical intervention continues to serve as the cornerstone therapeutic approach for breast cancer. While surgery is generally considered the preferred method for treating early-stage breast cancer in older women, regardless of comorbidity and frailty levels, PET may still be a viable alternative for those with significant frailty. In this particular patient cohort, no statistically significant difference in breast cancer-specific mortality was observed between surgery and PET arms (*p* = 0.251). Similarly, previous studies have reported no significant difference in breast cancer-specific survival when comparing surgery with PET (HR = 0.74, 95%CI: 0.40–1.37, *p* = 0.34) [39]. The similarity in outcomes may be influenced by the fact that mortality in this cohort is often due to non-cancer causes, such as cardiovascular or respiratory diseases, rather than breast cancer itself. Hence, while these results are intriguing, they should be interpreted with caution and further research is needed to validate PET as a comparably effective treatment strategy for surgery for frail older women with early-stage breast cancer.

Although PET is not recommended as the primary strategy for female patients aged over 70 years, 30% of older women with early-stage breast cancer still received PET in routine care. Patients who received PET as the initial treatment strategy had poorer physical functioning than those with surgery. However, after adjusting for relevant covariates, we found the risk gap of ACM between PET and surgery narrowed with increased HFRS and CCI levels. Notably, no statistical difference in BCSM was found between PET and surgery in frail patients. Overall, our study strengthens the national guideline in England [8] that surgery should be the first-line treatment for operable patients in good physical condition, irrespective of age, since surgery leads to better overall survival and is more cost-effective than PET [40]. Nevertheless, PET is a potentially effective strategy for older female patients who are physically unfit for surgery (frail or with a higher comorbidity level).

The prevailing preference for PET has its distinctive historical underpinnings. As endocrine therapy emerged and RCTs elucidated its clinical efficacy and effectiveness, PET garnered widespread adoption as a surgical alternative owing to its demonstrated non-inferiority and heightened efficiency relative to surgical interventions. Nowadays, although surgery is still recommended as the first-line treatment for patients who are physically fit for surgery, our study found that a high proportion of older female patients undergoing PET were either non-frail (69.6% of patients who received PET) or had low levels of CCI (70.2% of patients who received PET). These patients could have received surgery instead as their initial treatment. Instead, they received PET due to the challenges in evidence-based clinical decision-making and older female patients’ preference for treatment choices [41].

As mentioned in the National Audit of Older Women with Breast Cancer UK report [8], using PET as initial treatment for older female patients who are physically fit for surgery was inconclusive due to a lack of robust evidence to indicate the comparative effectiveness of PET versus surgery by different levels of physical functioning [42]. The findings from our study address this evidence gap and stratify the characteristics of patients who may benefit from PET to support clinical precision decision-making.

Several previous observational studies have evaluated the clinical effectiveness of surgery and PET [43,44,45,46]. Still, only one systematic review has summarised the treatment effects in older female patients with breast cancer from such observational studies [47]. This systematic review notably identified a potential source of bias in the observational studies, primarily stemming from confounding by indication [47]. Also, all the included studies evaluated the effects of tamoxifen rather than aromatase inhibitors; the latter is more often used in clinical practice for older female patients. Consequently, the findings presented in our study bear greater relevance to the contemporary standard of care for this demographic.

Our study identified the routine clinical practice of different types of PET given to older female patients. Although tamoxifen was still typically prescribed for patients with ER+ breast cancer, aromatase inhibitors (e.g., letrozole) were more prevalently prescribed for female patients aged ≥ 70 years and undergoing PET. A patient-level meta-analysis of four RCTs [48] evaluated the post-surgical adjuvant treatment effects between aromatase inhibitors and tamoxifen for postmenopausal women with ER+ early-stage breast cancer. The results demonstrated that aromatase inhibitors had superior effects on reducing the likelihood of recurrence (RR: 0.79, 95%CI: 0.69–0.90, *p* = 0.0005) and distant recurrence (RR: 0.83, 95% CI: 0.71–0.97; *p* = 0.018) [48]. Thus, the benefit of PET over surgery in very frail patients would be expected to be more marked.

In contrast to the existing body of literature, our study offers a comprehensive investigation into the relative clinical effectiveness of PET versus surgical interventions among elderly female patients, stratified across varying levels of physical functioning (using CCI and HFRS). Our research addresses a notable gap in the current knowledge landscape by leveraging population-based data. These findings constitute a valuable resource for clinicians seeking evidence-based insights to guide treatment optimisation for older women diagnosed with early-stage oestrogen-receptor-positive (ER+) breast cancer, particularly in varying levels of frailty and comorbidity.

This study had some strengths. Firstly, it evaluates the clinical efficacy of PET and surgical interventions utilising a high-quality UK primary care database and enriching its coverage through linkage with cancer registry records, secondary care data, socioeconomic indices, and death certificates. This extensive data integration strategy enhances the study’s sample size, assuring excellent statistical power (>99%). Secondly, the investigation encompasses two pivotal outcome measures to inform clinical decision-making: overall survival (OS) and breast cancer-specific mortality (BCSM). Furthermore, the sensitivity analysis considered competing risks, fortifying the results’ robustness and reliability.

Third, the bias by indication in patients undergoing PET and surgery was considered using the propensity score matching (calculated using IPTW to predict the likelihood of undergoing two treatments) and weighting in regression adjustment [35]. This doubly robust adjustment makes the survival outcomes comparable to the identical characteristics of two age strata by minimising the confounders (age, frailty, comorbidity, and socioeconomic status) between exposure and outcome [35]. Fourth, patients were stratified by using CCI and HFRS as proxies for frailty and comorbidities; hence, the results revealed the treatment effectiveness by age-related physical functioning. Finally, this study used aromatase inhibitors as the most common endocrine medication (55.5% of PET medications), which better reflects the current routine practice for early-stage breast cancer. Undeniably, there was still the limitation of the risks of residual bias from IPTW; however, according to the sensitivity analysis, the result may still indicate that PET is an appropriate treatment for older patients who are frail or comorbid. 

However, we acknowledge there are some limitations to this study. First, this study only selected three variables to match, and this may not perfectly reflect the influence on the survival consequences. There is some unmeasured confounding (e.g., diet habits, smoking and alcohol status, obesity, and psychological status) that may impact the survival benefits for the cohort [49]. Nonetheless, regression was used in this matched cohort study to adjust covariate variables (e.g., comorbidity and frailty) that can reflect the impact of such unmeasured confounding as smoking, alcohol status, and obesity on physical conditions. 

Second, although frailty and comorbidity commonly represent physical functioning in clinical practice, physical functioning as a complicated physical status may not entirely and accurately be quantified using only two indicators. Physical functioning also includes the body’s movement ability and quality of life [50]. In addition, although frailty or comorbidity was estimated in this study using the published validated algorithm [30,31], the frailty and comorbidity status for the patients in the real world may not be accurately quantified through the limited data source due to other factors that may determine frailty or comorbidity in routine practice. Therefore, this study only analysed the results based on the patients with the represented frailty or comorbidity status to inform the decision-making of treatments in older female patients. Future research on treatment effectiveness in older patients should assess survival outcomes based on their physical conditions, including frailty (HFRS) or comorbidity (CCI) scores.

Finally, this study defined PET and surgery as the initial treatment strategies, irrespective of the subsequent treatment received. Some older female patients who initially received PET would also receive surgery when their breast cancer progressed. However, post-progression management is a different clinical decision problem than early-stage breast cancer management upon diagnosis. To minimise misclassification bias in this study, patients who received PET and then received surgery after one year were excluded from the study cohort. 

In forthcoming research endeavours, it is imperative to undertake cost-effectiveness analyses contrasting PET with surgical interventions for frail older female patients. These analyses are pivotal in augmenting the precision of breast cancer management decisions. However, only one investigation conducted by Holmes et al. [51], derived from the Bridging the Age Gap in Breast Cancer study, has revealed that, except for a select subgroup comprising patients aged 90 years or older with a comorbidity score of 2 or 3, regardless of nodal status, surgical interventions manifest superior cost-effectiveness compared to PET [51]. Therefore, there is a pressing need for further assessments, stratified by frailty levels, to elucidate nuanced treatment selection paradigms. 

Furthermore, it is worth noting that countries beyond the United Kingdom, including the United States, Canada, and various European nations such as the Netherlands and Sweden, confront analogous dilemmas when determining the optimal therapeutic course for older female breast cancer patients [52]. In light of this, a global perspective should be adopted to facilitate comprehensive investigations aimed at discerning the specific patient characteristics that might derive maximal benefit from PET. Lastly, in the context of geriatric healthcare, where RCTs may not always be feasible or biased by indications, the judicious utilisation of propensity score matching or weighting represents a promising approach for evaluating treatment efficacy and effectiveness in other specific geriatric diseases.

## 5. Conclusions

Despite surgical procedures being the favoured effective initial treatment strategy for older elderly patients with primary breast cancer, its benefits on overall survival in patients with high levels of frailty are inferior to those of PET. Notably, given no substantial difference in the risk of non-cancer mortality between PET and surgery in the frail group, PET could be considered a suitable initial treatment option for older patients who are unsuitable for surgery due to frailty or comorbidity. To prove this, future research is recommended to include life expectancy estimates, measure patients’ physical conditions specified by frailty (HFRS) or comorbidity (CCI) scores, and further investigate the cost-effectiveness of PET to inform evidence-based healthcare decision-making.

## Figures and Tables

**Figure 1 cancers-16-00749-f001:**
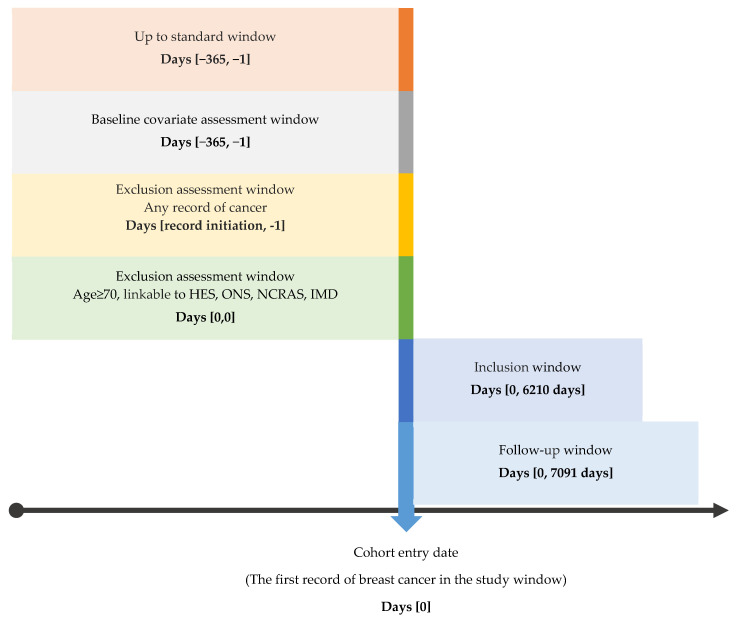
Diagram of the cohort study design.

**Figure 2 cancers-16-00749-f002:**
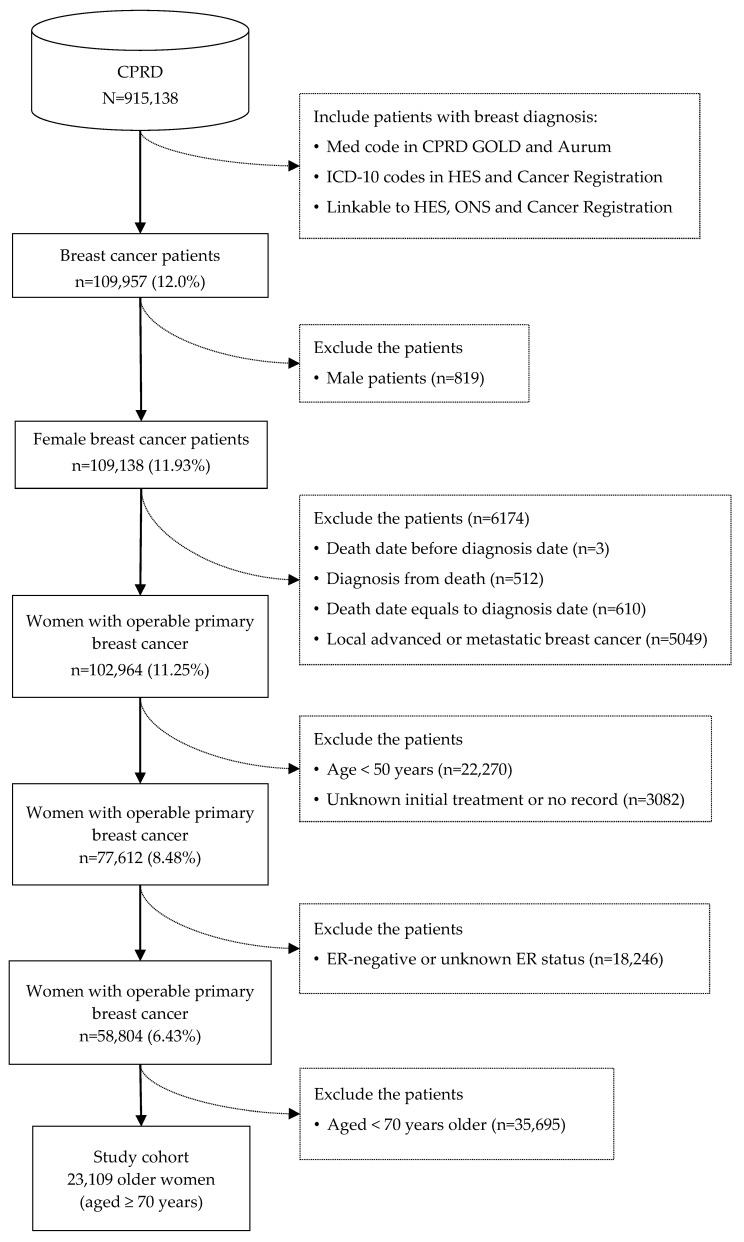
Study cohort identification process.

**Figure 3 cancers-16-00749-f003:**
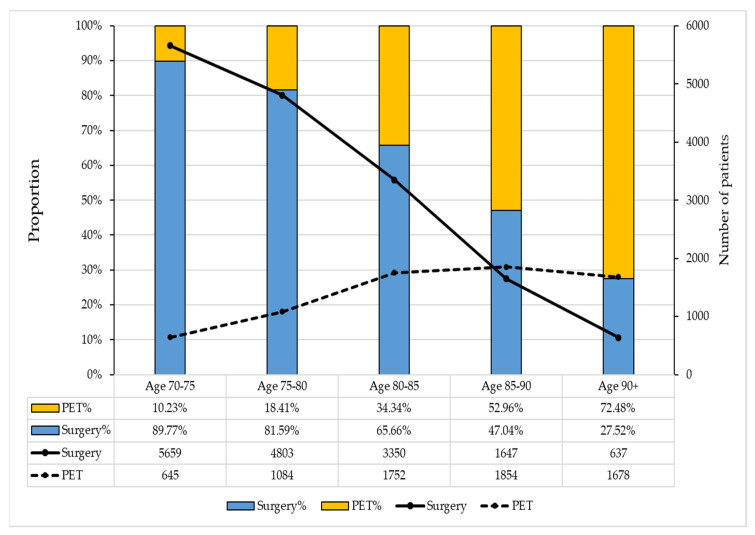
Number and proportion of patients undergoing initial treatment at different age groups. PET: Primary endocrine therapy.

**Figure 4 cancers-16-00749-f004:**
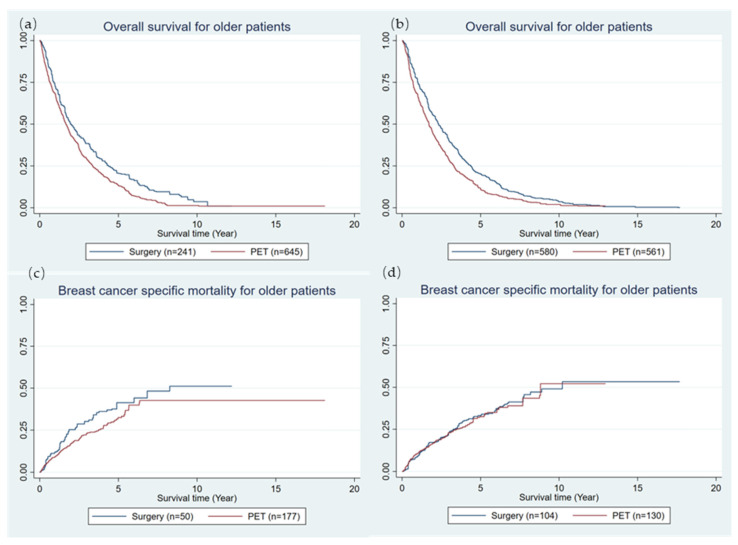
Overall survival and breast cancer-specific mortality curves between surgery and PET in older female patients with high-level HFRS and CCI. HFRS: hospital frailty risk score; CCI: Charlson comorbidity index. (**a**,**c**) High level of HFRS; (**b**,**d**) High level of CCI.

**Table 1 cancers-16-00749-t001:** Summary of the study databases.

Source	Dataset	Available Duration	Population
National Cancer Registration and Analysis Services (Set 18)	Cancer Registry	January 1990 to December 2016	England
	Systemic Anti-Cancer Treatment	January 2014 to September 2017	England
	National Radiotherapy Dataset	April 2012 to September 2017	England
Clinical Practice Research Datalink (Set 19)	CPRD Gold	1987 to June 2020 (Set 19)	UK
	CPRD Aurum	1995 to June 2020 (Set 19)	England
Hospital Episode Statistics (Set 18)	Admitted Patient Care	April 1997 to June 2019 (Set 18)	England
	Outpatient	April 2003 to June 2019 (Set 18)	England
Office for National Statistics (Set 18)	Death registration	January 1998 to May 2019	England
	Index of Multiple Deprivation	April 2007 to November 2015	England

(Note) SACT: Systemic Anti-Cancer Treatment; RTDS: National Radiotherapy Dataset; HES APC: Hospital Episode Statistics Admitted Patient Care. The available duration was according to the version of set 18 (released in April 2020) and set 19 (released in October 2020).

**Table 2 cancers-16-00749-t002:** Characteristics of the study cohort.

	Factor	Surgery (n = 16,096)	PET (n = 7013)	*p* Value *	SMD **
**Age**	Median (IQR)	77 (73–82)	85 (80–89)	<0.001	-
	Mean (SD)	77.8 (5.9)	84.3 (6.8)	<0.001	-
**Follow-up (year)**	Median (IQR)	6.49 (3.72–10.16)	2.77 (1.15–5.10)	<0.001	-
	Mean (SD)	7.23 (4.41)	3.73 (3.51)	<0.001	-
**CCI**	Median (IQR)	2 (2–3)	3 (2–4)	<0.001	-
	Low (0–2)	80.4%	70.2%	<0.001	0.002
	Intermediate (3–4)	16.0%	21.8%	<0.001	0.002
	High (≥5)	3.6%	8.0%	<0.001	−0.008
**HFRS**	Median (IQR)	2.9 (1.5–5.4)	5.5 (2.3–11.2)	<0.001	-
	Non-frail (0–5)	51.2%	12.9%	<0.001	0.013
	Pre-frail (6–15)	9.1%	21.2%	<0.001	−0.012
	Frail (≥15)	1.5%	9.2%	<0.001	−0.014
	Missing	38.2%	56.7%	<0.001	0.007
**IMD**	Median (IQR)	5 (2–7)	5 (3–8)	0.129	-
	1–2	20.6%	18.7%	0.001	0.051
	3–4	18.4%	17.6%	0.182	0.014
	5–6	16.0%	16.3%	0.482	−0.006
	7–8	7.8%	7.9%	0.681	−0.019
	9–10	10.9%	14.5%	<0.001	−0.042
	Missing	26.5%	25.0%	0.006	−0.059

IQR: Interquartile range; SD: standard deviation; ST.diff: standardised mean difference; CCI: Charlson comorbidity index; HFRS: hospital frailty risk score; IMD: index of multiple deprivations. * The continuous variable (age, median value of CCI, HFRS, and IMD) used Wilcoxon signed-rank tests, and the categorical variable (levels of CCI, HFRS, and IMD) used the Chi-square test. ** SMD: The difference of standardised mean value for the inverse probability of treatment weights estimated using the propensity score. SMD < 0.1 means the variables were well controlled.

**Table 3 cancers-16-00749-t003:** Tumour characteristics of the study cohort.

Factors		Surgery (n = 16,096)	PET (n = 7013)
**Tumour Grade**	G1	2380 (14.79%)	588 (8.38%)
	G2	7895 (49.05%)	1918 (27.35%)
	G3	3224 (20.03%)	476 (6.79%)
	GX	1033 (6.42%)	1921 (27.39%)
	Missing	1564 (9.72%)	2110 (30.09%)
**NPI**	Median (IQR)	3.6 (3.24–4.5)	3.3 (2.4–3.6)
	I	73 (0.45%)	5 (0.07%)
	II	434 (2.70%)	0 (0.00%)
	III	1039 (6.46%)	7 (0.10%)
	IV	2057 (12.78%)	9 (0.13%)
	V	543 (3.37%)	1 (0.01%)
	Missing	11,950 (74.24%)	6991 (99.69%)
**HER-2 status**	Positive	553 (3.44%)	132 (1.88%)
	Negative	3970 (24.66%)	945 (13.47%)
	Unknown	11,573 (71.90%)	5936 (84.64%)
HER-2 and ER/PR	Positive	553 (3.43%)	132 (1.88%)

(Note) PET: primary endocrine therapy; IQR: Interquartile range; NPI: Nottingham prognosis.

**Table 4 cancers-16-00749-t004:** Factors associated with the risk of all-cause mortality derived from the propensity score weighted and adjusted Cox proportional hazard model.

	Hazard Ratio (95%CI)	*p* Value
**Age** (raw)	1.1 (1.1, 1.1)	<0.001
**Age** (time-varying)	1.0 (1.0, 1.0)	0.108
**Treatment of PET** (reference: surgery)	1.9 (1.8, 2.0)	<0.001
**Frailty** (reference: non-frail)		
Pre-frail level	1.4 (1.3, 1.5)	<0.001
Frail level	2 (1.9, 2.2)	<0.001
**CCI** (reference: low level)		
Intermediate level	1.3 (1.3, 1.4)	<0.001
High level	1.8 (1.7, 1.9)	<0.001
**IMD** (reference: IMD decile 1–2)		
3–4	1.0 (1.0, 1.1)	<0.001
5–6	1.0 (1.0, 1.1)	<0.001
7–8	1.1 (1.0, 1.2)	<0.001
9–10	1.2 (1.1, 1.3)	<0.001
No observations	1.2 (1.1, 1.3)	<0.001

IPTW: the inverse probability of treatment weight; CI: confidence interval; PET: primary endocrine therapy; HFRS: hospital frailty risk score; CCI: Charlson comorbidity index; IMD: index of multiple deprivation. Age (raw) means the analytical sets were not adjusted with time-varying; Age (time-varying) means the analytical sets were adjusted with time-varying.

**Table 5 cancers-16-00749-t005:** The mortality risk comparing primary endocrine therapy against surgery in three levels of frailty score and Charlson comorbidity index.

Group	All-Cause Mortality		Breast Cancer-Specific Mortality	
	HR (95%CI)	*p* Value	HR (95%CI)	*p* Value
** *Level of frailty* **				
Non-frail (HFRS: 0 to 5) (n = 19,327)	2.1 (2.0, 2.2)	<0.0001	3.0 (2.8, 3.2)	<0.0001
Pre-frail (HFRS: 6 to 15) (n = 2903)	1.9 (1.7, 2.1)	<0.0001	2.1 (1.7, 2.5)	<0.0001
Frail (HFRS ≥ 15) (n = 879)	1.2 (1.1, 1.5)	0.006	1.2 (0.9, 1.9)	0.251
** *Level of comorbidity* **				
Low level of CCI (n = 17,863)	2.1 (2.0, 2.2)	<0.0001	3.0 (2.8, 3.2)	<0.0001
Intermediate level of CCI (n = 4101)	1.9 (1.7, 2.1)	<0.0001	2.2 (1.9, 2.6)	<0.0001
High level of CCI (n = 1145)	1.4 (1.2, 1.7)	<0.0001	1.5 (1.1, 2.1)	0.015

HR: hazard ratio; CI: confidence interval; HFRS: hospital frailty risk score; CCI: Charlson comorbidity index.

**Table 6 cancers-16-00749-t006:** Competing risk regression of PET compared to surgery by levels of frailty and comorbidity.

Group	Model 1		Model 2	
	SHR (95% CI)	*p* Value	SHR (95% CI)	*p* Value
** *Level of frailty* **				
Non-frail (0–5)	2.2 (2.1, 2.3)	<0.0001	1.2 (1.1, 1.3)	<0.0001
Pre-frail (6–15)	1.9 (1.7, 2.1)	<0.0001	1.4 (1.2, 1.5)	<0.0001
Frail (≥15)	1.2 (1.0, 1.4)	0.082	1.1 (0.9, 1.4)	0.261
** *Level of comorbidity* **				
Low level of CCI	2.2 (2.1, 2.3)	<0.0001	1.2 (1.1, 1.2)	<0.0001
Intermediate level of CCI	1.9 (1.7, 2.1)	<0.0001	1.4 (1.2, 1.5)	<0.0001
High level of CCI	1.4 (1.2, 1.6)	<0.0001	1.1 (0.9, 1.4)	0.093

HR: hazard ratio; SHR: subdistribution hazard ratio; CI: confidence interval; HFRS: hospital frailty risk score; CCI: Charlson comorbidity index; BCSM: breast cancer-specific mortality; ACM: all-cause mortality; Model 1 compared the competing risk between ACM and BCSM; Model 2 compared the competing risk between other causes of death and BCSM.

## Data Availability

In this study, we used anonymised patient-level data from the CPRD that are not publicly available due to confidentiality considerations. However, researchers can access CPRD’s databases by contacting the MHRA. Details of the application process and conditions of access are available at https://www.cprd.com/Data-access (accessed on 21 June 2021). This study is based on data from the Clinical Practice Research Datalink obtained under licence from the UK Medicines and Healthcare Products Regulatory Agency (MHRA). The data are provided by patients and collected by the NHS as part of their care and support. Hospital Episode Statistics (HES) and Office of National Statistics mortality data are subject to Crown copyright (2023) protection and re-used with the permission of The Health and Social Care Information Centre, with all rights reserved. The study protocol was approved by CPRD’s Independent Scientific Advisory Committee (ISAC) (reference: 20_079R). The OPCS Classification of Interventions and Procedures, codes, terms, and text are Crown copyright (2016) published by Health and Social Care Information Centre, also known as NHS Digital and licensed under the Open Government Licence available at www.nationalarchives.gov.uk/doc/open-government-licence/opengovernment-licence.htm (accessed on 21 June 2021). NCRAS Data were supplied with permission from Public Health England (2020). The interpretation and conclusions contained in this study are those of the authors alone and not necessarily those of the MHRA, NIHR, Public Health England, or the Department of Health and Social Care.

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
