# Peer review of "Survival Outcomes in Older Women with Oestrogen-Receptor-Positive Early-Stage Breast Cancer: Primary Endocrine Therapy vs. Surgery by Comorbidity and Frailty Levels"

_cancers, 2024, doi:10.3390/cancers16040749_

Round 1

Reviewer 1 Report

Comments and Suggestions for Authors

This well-written manuscript is interesting. The authors have studied and compared the place of primary endocrine therapy to surgery in elderly and very frail breast cancer patients.

I have no comments concerning the methodology and statistical analysis . I suggest that in the conclusion and for the design of future needed studies that the authors insist on mentioning life expectancy and inclusion of  fragility scores such as HHRS (Hospital Frailty Risk Score) or (CCI) The Charlson Comorbidity Index)

Author Response

[Reply] We sincerely appreciate the reviewer's positive remarks and thoughtful evaluation of our manuscript. Your feedback has been invaluable in refining our work. We have diligently considered your comments to enhance the clarity and quality of the manuscript. In response to your suggestions, we have emphasised the key points and revised the conclusion section. These changes are now highlighted in yellow on the manuscript for your convenience.

[Lines 457-459]

"Future research on treatment effectiveness in older patients should assess survival outcomes based on their physical conditions, including frailty (HFRS) or comorbidity (CCI) scores."

[Lines 487-495]

"Despite surgical procedures being the favoured, effective initial treatment strategy for older elderly patients with primary breast cancer, its benefits on overall survival in patients with high levels of frailty are inferior to those of PET. Notably, given no substantial difference in the risk of non-cancer mortality between PET and surgery in the frail group, PET could be considered a suitable initial treatment option for older patients who are unsuitable for surgery due to frailty or comorbidity. To prove this, future research is recommended to include life expectancy estimates, measure patients' physical conditions specified by frailty (HFRS) or comorbidity (CCI) score, and further investigate the cost-effectiveness of PET to inform evidence-based healthcare decision-making."

Reviewer 2 Report

Comments and Suggestions for Authors

Major

• Results and Conclusions: The introduction addresses the knowledge gap related to the comparative effectiveness of primary endocrine therapy versus surgery, specifically focusing on the risk of noncancer mortality in older female patients with varying degrees of physical functioning. The results indicate that the risk of noncancer mortality is not significantly different in patients with high frailty. However, the conclusion advocates the benefits of primary endocrine therapy, creating an inconsistency with the study's findings.

• Discussion: Despite the assertion that Primary Endocrine Therapy (PET) could be a potentially effective strategy for older female patients who are physically unfit for surgery (frail or with a higher comorbidity level), this statement lacks data-based support in this study.

• Frailty Analysis: While frailty is a pivotal concern, the analysis lacks quantitative depth. In Figure 4, panels (a) and (c) illustrate outcomes for older patients with elevated levels of HFRS. To enhance understanding, please provide outcomes for various threshold levels of high HFRS.

Minor

·         Abbreviations: Many abbreviations are used and it is not easy to follow the text. Please try to reduce the abbreviations, particularly, in the abstract.

Additional comment: Figure 2 is well-designed and it is easy to follow.

Comments on the Quality of English Language

fine.

Author Response

Major

[Comment 1]

Results and Conclusions: The introduction addresses the knowledge gap related to the comparative effectiveness of primary endocrine therapy versus surgery, specifically focusing on the risk of noncancer mortality in older female patients with varying degrees of physical functioning. The results indicate that the risk of noncancer mortality is not significantly different in patients with high frailty. However, the conclusion advocates the benefits of primary endocrine therapy, creating an inconsistency with the study's findings.

[Reply] We have revised the conclusion section to reflect the main finding and the implications to align with the introduction section.

[Lines 487-495]

"Despite surgical procedures being the favoured, effective initial treatment strategy for older elderly patients with primary breast cancer, its benefits on overall survival in patients with high levels of frailty are inferior to those of PET. Notably, given no substantial difference in the risk of non-cancer mortality between PET and surgery in the frail group, PET could be considered a suitable initial treatment option for older patients who are unsuitable for surgery due to frailty or comorbidity. To prove this, future research is recommended to include life expectancy estimates, measure patients' physical conditions specified by frailty (HFRS) or comorbidity (CCI) score, and further investigate the cost-effectiveness of PET to inform evidence-based healthcare decision-making."

[Comment 2]

Discussion: Despite the assertion that Primary Endocrine Therapy (PET) could be a potentially effective strategy for older female patients who are physically unfit for surgery (frail or with a higher comorbidity level), this statement lacks data-based support in this study.

[Reply] We have revised the conclusion section, and this sentence has also been modified. 

[Lines 487-495]

"Despite surgical procedures being the favoured, effective initial treatment strategy for older elderly patients with primary breast cancer, its benefits on overall survival in patients with high levels of frailty are inferior to those of PET. Notably, given no substantial difference in the risk of non-cancer mortality between PET and surgery in the frail group, PET could be considered a suitable initial treatment option for older patients who are unsuitable for surgery due to frailty or comorbidity. To prove this, future research is recommended to include life expectancy estimates, measure patients' physical conditions specified by frailty (HFRS) or comorbidity (CCI) score, and further investigate the cost-effectiveness of PET to inform evidence-based healthcare decision-making."

[Comment 3]

Frailty Analysis: While frailty is a pivotal concern, the analysis lacks quantitative depth. In Figure 4, panels (a) and (c) illustrate outcomes for older patients with elevated levels of HFRS. To enhance understanding, please provide outcomes for various threshold levels of high HFRS.

[Reply] In light of the study's focus and the journal's word count constraints, we have included the results and figures for different HFRS and CCI threshold levels in the supplementary materials (Appendix 13).

[Line 293-296]

“When stratifying patients by levels of frailty and comorbidity (details in Supplementary file Appendix 13), non-frail patients who underwent surgery had a significantly longer OS time (median: 6.8 years; IQR: 4.0-10.6) compared to their matched PET counterparts (median: 3.2 years; IQR: 1.4-6.0).”

[Comment 4]

Abbreviations: Many abbreviations are used, and it is not easy to follow the text. Please try to reduce the abbreviations, particularly, in the abstract.

[Reply] We revised the abstract to minimise the use of abbreviations.

Additional comment:

Figure 2 is well-designed and it is easy to follow.

[Reply] Thank you for your positive feedback on Figure 2. Your acknowledgement is greatly appreciated.

Round 2

Reviewer 2 Report

Comments and Suggestions for Authors

The authors responded sufficiently to the comements.

Thank you.